# Feasibility of a space-borne terahertz heterodyne spectrometer for atomic oxygen and temperature in the mesosphere and lower thermosphere

Peder Bagge Hansen<sup>1,2</sup>, Martin Wienold<sup>1</sup>, and Heinz-Wilhelm Hübers<sup>1,2</sup>

<sup>1</sup>German Aerospace Center (DLR), Institute of Optical Sensor Systems, Berlin, Germany

<sup>2</sup>Humboldt-Universität zu Berlin, Department of Physics, Berlin, Germany

Correspondence: Peder Bagge Hansen (peder.hansen@dlr.de)

**Abstract.** We investigate the feasibility of a satellite-borne heterodyne spectrometer for the retrieval of atomic oxygen concentration and temperature in the mesosphere and lower thermosphere. We use the vertical density and temperature profiles provided by the NRLMSIS 2.1 atmosphere model to simulate 2.1 THz and 4.7 THz atomic oxygen emission spectra as measured by a satellite in a near polar circular orbit at 500 km altitude. We then apply retrieval algorithms for the atomic oxygen concentration and temperature and compare the retrieved profiles to the reference, i.e. the original NRLMSIS 2.1 profiles. The emission spectra are simulated using radiative transfer under the assumption of local thermodynamic equilibrium.

By considering two separate heterodyne receivers with sensitivity of  $11,000\,\mathrm{K}$  and  $25,000\,\mathrm{K}$  noise temperature for the  $2.1\,\mathrm{THz}$  and  $4.7\,\mathrm{THz}$  lines, respectively, and data accumulated over 177 seconds of measurement time, corresponding to a ground track of  $1,250\,\mathrm{km}$ , we can retrieve vertical temperature profiles from  $100\,\mathrm{km}$  altitude to  $200\,\mathrm{km}$  altitude within  $\pm 2\%$  relative uncertainties and an atomic oxygen concentration profile from  $110\,\mathrm{km}$  to  $300\,\mathrm{km}$  within  $\pm 3\%$  relative uncertainties. From  $100\,\mathrm{km}$  to  $110\,\mathrm{km}$  the uncertainty in the atomic oxygen concentration is higher but still within  $\pm 15\%$ .

#### 1 Introduction

Atomic oxygen plays a critical role in the mesosphere and lower thermosphere (MLT) due to its influence on atmospheric dynamics, chemical processes, and energy transfer mechanisms (Mlynczak and Solomon, 1993; Riese et al., 1994). However, its abundance and variability are not yet well understood, with current estimates largely derived from indirect measurements. These indirect measurements involve the abundances of OH, O<sub>2</sub>, or O<sub>3</sub> and rely on photo-chemical reaction models (Mlynczak et al., 2018; Sheese et al., 2011). Direct measurements of atomic oxygen can be achieved through its terahertz (THz) fine structure transitions at 2.1 THz and 4.7 THz corresponding to the transitions from the state <sup>3</sup>P<sub>0</sub> to <sup>3</sup>P<sub>1</sub> and <sup>3</sup>P<sub>1</sub> to <sup>3</sup>P<sub>2</sub>, respectively. Because these transitions are in local thermal equilibrium (LTE) (Sharma et al., 1994), they provide a more accurate measure of atomic oxygen concentrations from the observed emissions, without needing complex assumptions about the state of the atmosphere. Satellite observations using limb geometry improve the retrieval process by allowing systematic analysis of the atmosphere at different tangential altitudes.

The Cryogenic Infrared Spectrometers and Telescopes for the Atmosphere (CRISTA) on board of the space shuttle Atlantis and afterwards Discovery, has measured atomic oxygen at  $4.7\,\mathrm{THz}$  in the MLT (Offermann et al., 1999; Grossmann et al., 2000). However, these measurements did not resolve the spectral lines, providing only concentration estimates at altitudes above  $130\,\mathrm{km}$  with an uncertainty of  $\pm 25\,\%$ , where the density is lower and thus lower absorption is expected. The ability to resolve line profiles is crucial for probing lower atmospheric layers where much of the information lies in the line wings due to a saturation of the signals. By resolving the line profiles, neutral winds can also be derived from the resulting Doppler shifts (Wu et al., 2016). The ratio between the  $2.1\,\mathrm{THz}$  and  $4.7\,\mathrm{THz}$  emissions is related to the thermal conditions of the atmosphere. Thus, the measurement of both transitions facilitates temperature retrieval.

The first spectrally resolved measurements of atomic oxygen fine structure transitions from the atmosphere were accomplished with the GREAT (German Receiver for Astronomy at Terahertz Frequencies) heterodyne spectrometer on board of the SOFIA (Stratospheric Observatory for Infrared Astronomy) airplane (Richter et al., 2021) and, more recently, by the Oxygen Spectrometer for Atmospheric Science on a Balloon (OSAS-B) which is a heterodyne spectrometer on a stratospheric balloon (Wienold et al., 2024). The GREAT receiver also made the first detection of <sup>18</sup>O in the MLT, confirming that the atmosphere of Earth contains a higher fraction of the <sup>18</sup>O isotope than ocean water, i.e. the Dole effect (Wiesemeyer et al., 2023).

Keystone, which is a satellite mission concept for measuring atomic oxygen in the MLT, has recently entered a phase 0 study (KEY, 2024). The Keystone mission would utilise state-of-the-art heterodyne spectroscopy technology to measure the 2.1 THz and 4.7 THz transitions, enabling the retrieval of both atomic oxygen concentrations and temperature profiles.

- In this study, we investigate the feasibility of a satellite mission designed to retrieve atomic oxygen and temperature vertical profiles in the MLT from the atomic oxygen fine structure transitions at 2.1 THz and 4.7 THz. Our approach extends the work of previous feasibility studies by Sharma et al. (Zachor and Sharma, 1989; Sharma et al., 1990) by incorporating state-of-the art heterodyne spectroscopy technology. Additionally, we account for spatial variations in atmospheric properties along the satellite's path and consider the impact of atmospheric winds.
- To simulate mission data, we use the atomic oxygen and temperature values from the Naval Research Laboratory Mass Spectrometer and Incoherent Scatter model (NRLMSIS 2.1) (Emmert et al., 2022) along with winds from the Horizontal Wind Model (HWM14) (Drob et al., 2015). We then apply retrieval algorithms to obtain vertical profiles of atomic oxygen and temperature, and then compare these profiles to the reference NRLMSIS 2.1 profiles to evaluate the mission's feasibility.

This study aids the development of future missions for high-resolution measurements of atomic oxygen and temperature in the MLT region, such as the Keystone mission.

# 2 Mission Scenario

The instrument for measuring the 2.1 THz and 4.7 THz emissions is assumed to consist of two separate receivers which share the same optical frontend using a dichroic beamsplitter to separate the two frequency bands. Consequently, the two transitions are measured simultaneously having the same integration times and viewing angles. For receiving the THz emissions, a lens or mirror with a diameter of  $40\,\mathrm{cm}$  is assumed. According to a diffraction limited system, the FWHM of the field of view is then

 $0.025^{\circ}$  and  $0.011^{\circ}$  for the  $2.1\,\mathrm{THz}$  and  $4.7\,\mathrm{THz}$  receiver, respectively. For a  $500\,\mathrm{km}$  orbit altitude, this results in a  $1.0\,\mathrm{km}$  field of view (FWHM) at the  $100\,\mathrm{km}$  tangential point for the  $2.1\,\mathrm{THz}$  receiver and  $0.45\,\mathrm{km}$  for the  $4.7\,\mathrm{THz}$  receiver.

The instrument is assumed to have a frequency stabilised local oscillator with a line width that is negligible compared to the Doppler widths of the transitions which are 5.2 MHz (FWHM) and 12.0 MHz (FWHM) for the 2.1 THz and 4.7 THz transition, respectively, at 200 K. The receivers are radio-metrically calibrated by blackbody sources included as part of the instrument.

For the receivers, single-side-band (SSB) noise temperatures of  $11,000\,\mathrm{K}$  and  $25,000\,\mathrm{K}$  are assumed at  $2.1\,\mathrm{THz}$  and  $4.7\,\mathrm{THz}$ , respectively. The receivers would be based on planar Schottky diode mixer technology. Such receivers have been demonstrated at  $2.5\,\mathrm{THz}$  (Gaidis et al., 2000) and  $2.1\,\mathrm{THz}$  (Siles et al., 2024) with SSB noise temperatures of  $16,500\,\mathrm{K}$  and  $17,000\,\mathrm{K}$ , respectively. Assuming some improvement of the noise temperature and a linear scaling of noise temperature with frequency we arrive at the assumed noise temperature. For a spectral bin width of  $1\,\mathrm{MHz}$  and one second integration time, the root-mean-squared (RMS) noise level is  $11.0\,\mathrm{K}$  and  $25.0\,\mathrm{K}$  in units of brightness temperature, respectively. For an arbitrary integration time,  $t_{int}$ , the noise scales with  $1/\sqrt{t_{int}}$  within the Allen time of the receiver. It is assumed that the Allen time of the receiver is larger than the integration times used in this study. The assumed receiver parameters are summarised in Table 1.

**Table 1.** Parameters of the two heterodyne receivers.  $T_{sys}$  is the noise temperature, B is the spectral bin width, and  $\Delta T$  is the resulting noise for an integration time ( $t_{int}$ ) of one second.

| Receiver | T <sub>sys</sub> / K | B / MHz | t <sub>int</sub> / s | $\Delta T/K$ |
|----------|----------------------|---------|----------------------|--------------|
| 2.1 THz  | 11,000               | 1       | 1                    | 11.0         |
| 4.7 THz  | 25,000               | 1       | 1                    | 25.0         |

For simplicity, the Earth is modeled as a sphere with a radius of 6,371 km. The satellite moves on a polar circular orbit at 500 km altitude and an inclination with the equator of 97.5°. The orbit parameters are summarised in Table 2. The satellite trajectory is arbitrarily chosen to cross the zero latitude and zero longitude at the date and time 2022-09-07, 10:00 UTC.

**Table 2.** Orbit parameters for the polar circular orbit. The inclination is given by the angle with the equator.

| Altitude / km              | 500   |
|----------------------------|-------|
| Inclination / deg          | 97.5  |
| Orbit velocity / km/s      | 7.62  |
| Ground speed / km/s        | 7.06  |
| Orbital period / min       | 94.47 |
| No. of revolutions per day | 15.24 |

To retrieve a single vertical atomic oxygen or temperature profile, multiple line-of-sight (LOS) measurements are required, with each LOS intersecting the atmosphere at a different tangential height. The set of measurements for a retrieval of a single

vertical profile will be referred to as a "scan". In this study, we focus on retrieval above  $100\,\mathrm{km}$  and thus we set the minimum tangential height at  $100\,\mathrm{km}$  altitude. The maximum, which is related to the retrieval scheme (discussed later), is set at  $311\,\mathrm{km}$  altitude. Each scan includes measurements at 45 different tangential heights, resulting in a total of 90 measurements ( $2 \times 45$ ) of the  $2.1\,\mathrm{THz}$  and  $4.7\,\mathrm{THz}$  transitions. All measurements are done in the direction of the orbit track (i.e. along track).

A time-budget of 177s is allocated for a single scan. For the satellite at  $500 \,\mathrm{km}$  altitude, this corresponds to a ground track of  $1250 \,\mathrm{km}$ . After each measurement at a specific tangential height, we allocate 0.5s to adjust the optics to a new tangential height. Additionally, 10s are reserved for a radiometric calibration at the start of each scan. This leaves 144.5s ( $177s - 0.5s \times 45 - 10s$ ) of integration time to be distributed among the measurements in a scan. Figure 1 illustrates how the integration time is distributed across the different tangential height measurements in a scan. The integration time is adjusted based on the varying sensitivity of the measured spectra to changes in atomic oxygen concentration and temperature. For example, longer integration times are used for the lowest tangential height measurements, where stronger self-absorption is expected, and much of the information lies in the line wings, which require a high signal-to-noise ratio. Similarly, longer integration times are allocated at higher tangential height measurements, where lower concentrations and thus weaker signals are expected. At  $100 \,\mathrm{km}$  altitude, measurements are obtained for tangential heights at every  $1 \,\mathrm{km}$  allowing for a high vertical resolution. At higher altitudes, a slower variation of the atomic oxygen and temperature with altitude is expected and the tangential height sampling is decreased.

**Figure 1.** The tangential heights and the corresponding integration times for the measurements in a scan. Measurements are performed at 45 different tangential heights.

# 3 Simulation of Mission Data

#### 3.1 Radiative Transfer

90

The radiative transfer of the spectral radiance from the 2.1 THz and 4.7 THz emission lines is done with the assumption of LTE. By further neglecting scattering, the radiative transfer equation (RTE) along a one-dimensional LOS takes the form of

$$\frac{dI_{\nu}}{ds} = \epsilon_{\nu} - \frac{\epsilon_{\nu}}{B_{\nu}} \cdot I_{\nu} \,, \tag{1}$$

where  $I_{\nu}$  is the spectral radiance in  $Wm^{-2}sr^{-1}Hz^{-1}$ , ds is a differential path length,  $\epsilon_{\nu}$  is the emission coefficient, and  $B_{\nu}$  is the blackbody spectral radiance. For solving equation (1), we discretise the atmosphere into a series of consecutive spherical shells, c.f. Figure 2. We then assume a homogeneous temperature and atomic oxygen concentration along the LOS inside the spherical shells. The solution for a LOS through the i'th shell is

$$I_{\nu}(\mathbf{s}_{i+1}) = B_{\nu} + (I_{\nu}(\mathbf{s}_i) - B_{\nu}) \cdot \exp\left(-\frac{\epsilon_{\nu}}{B_{\nu}} \cdot \Delta s\right) , \qquad (2)$$

where  $I_{\nu}(s_{i+1})$  is the spectral radiance at the position of  $s_{i+1}$ , i.e. after propagating the radiance through the i'th spherical shell, and  $I_{\nu}(s_i)$  is the spectral radiance upon entering the i'th spherical shell at position  $s_i$ .  $\Delta s$  is the distance of the LOS inside the spherical shell. In order to evaluate equation (2),  $B_{\nu}$  is calculated by the temperature value and  $\epsilon_{\nu}$  is calculated by the values of the temperature, the atomic oxygen concentration, and the winds in the atmosphere. The values at the point half way through the spherical shells  $((s_{i+1} + s_i)/2)$  are used. Winds are entering the RTE by projecting the 3D wind vector along the LOS and incorporating the corresponding Doppler shift into the line-profile. The line-profile is contained in the emission coefficients  $(\epsilon_{\nu})$ . Due to the low pressure at the considered altitudes, collisional broadening is neglected and the line-profile is given by a Gaussian function with a width corresponding to the temperature at the considered shell. Thus, the emission coefficient in SI units for either of the transitions can be written as

$$\epsilon_{\nu}(\nu) = \frac{h\nu}{4\pi} \cdot \rho \cdot A_{lu} \frac{g_u \cdot \exp\left(\frac{E_u}{k_b \cdot T}\right)}{Z} \cdot P_{\nu}(\nu, \sigma, \nu_c) , \qquad (3)$$

where  $\nu$  is the frequency, h Planck's constant,  $\rho$  is the atomic oxygen number density,  $A_{lu}$  the Einstein coefficient for spontaneous emission,  $g_u$  the upper level degeneracy,  $E_u$  the upper level energy,  $k_b$  the Boltzmann constant, T the temperature, Z the electronic partition function, and  $P_{\nu}$  the Doppler profile. The width,  $\sigma$ , and central frequency,  $\nu_c$ , of the Doppler profile are given by

$$\sigma = \sqrt{\frac{k_b T}{mc^2}} \cdot \nu_0$$
, and (4)

$$\nu_c = \sqrt{\frac{c - w_{LOS}}{c + w_{LOS}}} \cdot \nu_0 \tag{5}$$

where m is the mass of atomic oxygen, c the speed of light in vacuum,  $\nu_0$  the rest frequency of the transition, and  $w_{LOS}$  the wind speed projected along the LOS. The atomic constants for evaluating the emission coefficients are summarised in Table 3.

**Table 3.** Parameters of the atomic oxygen transitions used in the radiative transfer.  $v_0$  refers to the rest frequency of the transitions,  $E_u$  refers to the upper level energy,  $g_u$  refers to the upper level degeneracy, and  $A_{lu}$  refers to the Einstein coefficients for spontaneous emission. Rest frequency of the 4.7 THz from Zink et al. (1991). Remaining values from the NIST database (Kramida et al., 2022).

| Transition                 | 2.1 THz              | 4.7 THz              |  |
|----------------------------|----------------------|----------------------|--|
| $v_{\theta}$ / THz         | 2.060                | 4.745                |  |
| $E_u$ / eV                 | 0.0281416            | 0.0196224            |  |
| $g_u$                      | 1                    | 3                    |  |
| $A_{lu}$ / s <sup>-1</sup> | $1.75 \cdot 10^{-5}$ | $8.91 \cdot 10^{-5}$ |  |

**Figure 2.** Schematic of a LOS intersecting a spherical shell in a discretised atmosphere.

# 3.2 Simulation of Spectra

Because of the large lens/mirror diameter and consequently the small beam divergence, the light collection of the instrument is approximated by a single one-dimensional LOS. Thus, a spectrum is simulated by propagating the one-dimensional LOS radiance shell by shell (equation (2)) according to the path of the LOS through the discretised atmosphere. We use a shell thickness of  $0.25\,\mathrm{km}$  up to an altitude of  $200\,\mathrm{km}$ . Above  $200\,\mathrm{km}$  altitude, an increasingly coarser discretisation is applied. At  $1000\,\mathrm{km}$ , the shell thickness is  $3\,\mathrm{km}$ . The boundary condition of zero radiance is used at altitudes larger than  $1000\,\mathrm{km}$ .

For the temperature and atomic oxygen values, we use the NRLMSIS 2.1 atmospheric model, while wind values are obtained from the HWM14 wind model. Figure 3 illustrates global sample temperature and atomic oxygen concentrations from the NRLMSIS 2.1 model at an altitude of 121 km, while Figure 4 illustrates global sample winds from the HWM14 model. Both

**Figure 3.** Sample temperatures and atomic oxygen densities from the NRLMSIS 2.1 model at an altitude of 121km at the date and time 2022-09-07, 12:43:15 UTC.

models provide data at arbitrary longitudes, latitudes, and altitudes. Consequently, the mission data is simulated using an atmosphere that varies with location, i.e. without assuming spherical symmetry.

To simulate measured spectra, we add Gaussian noise with a variance corresponding to the noise temperatures of the receivers and the integration times (see Table 1 and Figure 1). Examples of simulated measurements at tangential heights of  $100\,\mathrm{km}$  and  $121\,\mathrm{km}$  can be seen in Figure 5. In total, 33 scans are simulated. This corresponds to the satellite moving approximately one orbit. The simulated satellite orbit and the center positions of the simulated scans are shown in Figure 6. We define the center position of a scan as the longitude and latitude of the average vector of the tangential points in the scan.

**Figure 4.** Sample meridonal and zonal winds from the HWM14 model at an altitude of  $121\,\mathrm{km}$  at the date and time 2022-09-07, 12:43:15 UTC.

Figure 5. Examples of simulated spectra from scan number one. (a)  $2.1\,\mathrm{THz}$  spectrum at tangential height  $100\,\mathrm{km}$ . (b)  $4.7\,\mathrm{THz}$  spectrum at tangential height  $120\,\mathrm{km}$ . (c)  $2.1\,\mathrm{THz}$  spectrum at tangential height  $121\,\mathrm{km}$ . (d)  $4.7\,\mathrm{THz}$  spectrum at tangential height of  $121\,\mathrm{km}$ . The frequency axes corresponds to the rest frequencies of the atomic oxygen transitions (see Table 3). The position of the satellite, the integration times  $(t_{int})$ , and the spectral noise, are annotated in the figures. The different noise in the  $2.1\,\mathrm{THz}$  and  $4.7\,\mathrm{THz}$  spectra reflects the different receiver sensitivities. The lower noise at  $100\,\mathrm{km}$  tangential heights reflects the higher integration at this tangential height.

**Figure 6.** Global map of the center positions of the simulated scans and the satellite trajectory. The center positions are calculated as the average vector at the tangential points of the LOS in the scans. The ground track distance between the scan centres is  $1250\,\mathrm{km}$  corresponding to a time budget of  $177\,\mathrm{s}$ . The scans are numbered from 1 to 33.

# 4 Retrieval Methods

In order to retrieve the atomic oxygen concentration and the temperature from the simulated spectra, the atomic oxygen and the temperature are represented using a mathematical model with some parameters, i.e. a parametrisation. In the following, the choice of parametrisation and the method for solving the inverse problem are presented.

500 Linear region B-spline region 450 Resulting parametrization Linear profile B-splines 400 350 Altitude / km 300 250 200 150 100 14 15 16 17 -5 0 Residuals / % Log( atomic oxygen concentration /  $m^{-3}$  )

**Figure 7. Left:** The temperature parameterisation of the global average temperature profile from the NRLMSIS 2.1 at the date and time 2022-07-18, 00:00 UTC. From 100 km to 175 km altitude, the parametrisation is done using third order B-spline bases. The individual basis functions can be seen. At 175 km altitude and above, the parametrisation uses a so called Bates-profile.

**Right:** Residuals between the average temperature profile from NRLMSIS 2.1 and the parametrisation shown to the right. The parametrisation can be seen to describe the NRLMSIS 2.1 data within one Kelvin.

**Figure 8. Left:** The parametrisation of the global average atomic oxygen from the NRLMSIS 2.1 at the time and date 2022-07-18, 00:00 UTC. The parametrisation is done for the logarithm of the atomic oxygen. From 100 km to 300 km altitude, the parametrisation uses a B-spline basis. The individual basis functions can be seen. At 300 km altitude, the parametrisation uses a linear function. Note the logarithmic x-axis.

**Right:** Residuals between the average atomic oxygen concentration profile from NRLMSIS 2.1 and its parametrisation. The parametrisation can be seen to describe the NRLMSIS 2.1 data within 1.4 percent concentration deviations from 100 km to 200 km altitude.

# 4.1 Vertical Temperature Parametrisation

In analogy to the NRLMSIS model, we use B-splines together with a so called Bates-profile for the vertical temperature parametrisation (Emmert et al., 2022; de Boor, 1978; Bates and Massey, 1959). From 100km altitude (i.e. the start of the retrieval) and up to an altitude of 175km, the temperature is parametrised using a linear combination of third order B-spline basis functions. At an altitude of 175km the vertical temperature parametrisation transitions into the Bates-profile. The Bates-profile is given by equation (6)

$$T_{Bates}(r) = T_{ex} - (T_{ex} - T_B) \cdot \exp\left(-\kappa \cdot (r - 175)\right) , \tag{6}$$

where  $T_{ex}$  is the exosphere temperature,  $T_B$  is the temperature at an altitude of  $175 \,\mathrm{km}$ ,  $\kappa$  is a shape parameter, and r is the altitude in km. The Bates-profile is an empirically derived formula that basically describes an exponential convergence from a reference temperature of  $T_B$  at  $175 \,\mathrm{km}$  to a constant temperature of  $T_{ex}$  at higher altitudes (Jacchia, 1965).

The temperature parametrisation is inspired and thus similar to what is used in the NRLMSIS 2.1 model. However, in this study the temperature is modeled directly, and not the reciprocal. Furthermore, the distances between the B-splines are smaller at lower altitudes and the transition into the Bates-profile is at a higher altitude (175 km versus 122.5 km in the NRLMSIS 2.1 model). The altitude of the B-spline knots are: 95, 100, 105, 110, 115, 123, 135, 151, 175, and 199 km. As an example, the vertical temperature parametrisation of the global average temperature from NRLMSIS 2.1, at the date and time 2022-07-18, 00:00 UTC (arbitrarily chosen), can be seen in Figure 7. From the figure it can be seen how the B-spline bases are scaled in order to fit the vertical temperature profile.

There are three parameters for the Bates-profile:  $T_{ex}$ ,  $T_B$ , and  $\sigma$ , and there are 10 parameters for the coefficients of the B-spline basis functions. However the coefficients for the first three (lowest altitude) B-splines are reduced to two by constraining the curvature at  $100 \,\mathrm{km}$  altitude to be zero. The zero curvature constraint is applied in order to constrain the B-spline basis function centred at  $95 \,\mathrm{km}$  which only slightly overlaps with the altitude domain of the parametrisation. The last three B-spline coefficients (highest altitudes) are fixed by  $C^2$ -continuity with the Bates-profile at  $175 \,\mathrm{km}$ . This results in nine free parameters for the vertical temperature parametrisation.

# 4.2 Vertical Atomic Oxygen Parametrisation

The parametrisation of the vertical atomic oxygen concentration profile follows a similar approach to that of temperature. However, due to the high variation in atomic oxygen across the altitude range, the parametrisation is applied to the logarithm of the concentration. From  $100\,\mathrm{km}$  altitude and up till  $300\,\mathrm{km}$  altitude, the parametrisation is done with a linear combination of third order B-spline basis functions. The altitude of the B-spline knots are: 94, 100, 106, 112, 120, 133, 152, 182, 228, 300, and 372 km. For altitudes above  $300\,\mathrm{km}$ , the parametrisation uses a linear function of the form  $y = a \cdot x + b$ . As an example, the parametrisation of the global averaged atomic oxygen concentration profile at the date and time 2022-07-18, 00:00 UTC is shown in Figure 8.

The vertical atomic oxygen parametrisation contains two parameters for the linear function, and 11 parameters for the coefficients of the B-spline basis functions. As for the temperature parametrisation, the coefficients for the first three B-spline

coefficients (lowest altitudes) are reduced to two by the zero curvature constraint at  $100\,\mathrm{km}$  altitude. The last three B-spline coefficients (highest altitudes) are fixed by  $C^2$ -continuity with the linear function at  $300\,\mathrm{km}$ . This results in nine free parameters for the vertical atomic oxygen parametrisation.

# 4.3 Correction for Spherical Asymmetry

The LOS in a scan span multiple longitudes and latitudes, as shown in Figure 9. This is a natural consequence of the limb-sounding geometry and the satellite's movement during a scan. To account for variations in the vertical profiles across different longitudes and latitudes, i.e. the spherical asymmetry of the atmosphere, two or more scans can be combined for the simultaneous retrieval and interpolation of two or more vertical profiles covering the region traversed by the LOS. This is the approach of the retrieval described in Livesey and Read (2000). In our study, three scans are combined to introduce a second-order correction to the vertical temperature and atomic oxygen parametrisations. Due to the forward-looking geometry, this correction can be applied with respect to a single variable, α, which is the angle with respect to the center of the combined scan (see Figure 9). The introduction of such a correction has the additional advantage that vertical profiles can be obtained at arbitrary positions in the vicinity of the center of the combined scan. Without the correction the retrieved profiles would correspond to some kind of averages over the regions traversed by the LOS. These regions are larger than the ground track distance of the scans c.f. Figure 9.

With the second order correction, the vertical profiles are given as a function of the altitude, r, and  $\alpha$  by

$$T(r,\alpha) = T_{sym}(r) \cdot \left(1 + \alpha \cdot T_{1,corr}(r) + \alpha^2 \cdot T_{2,corr}(r)\right)$$

$$\rho(r,\alpha) = \rho_{sym}(r) \cdot \left(1 + \alpha \cdot \rho_{1,corr}(r) + \alpha^2 \cdot \rho_{2,corr}(r)\right) ,$$
(7)

where  $T_{sym}(r)$  and  $\rho_{sym}(r)$  are the spherical vertical temperature and atomic oxygen concentration parametrisation, respectively (Figure 7 and Figure 8).  $T_{1,corr}(r)$  and  $\rho_{1,corr}(r)$  are the altitude dependent first order corrections and  $T_{2,corr}(r)$  and  $\rho_{2,corr}(r)$  are the second order corrections. The altitude dependence of  $T_{1,corr}(r)$ ,  $\rho_{1,corr}(r)$ ,  $T_{2,corr}(r)$ , and  $\rho_{2,corr}(r)$  are parametrised identically using six B-splines from  $100\,\mathrm{km}$  and until  $200\,\mathrm{km}$ . The altitude of the B-spline knots are: 77, 100, 123, 155, 200, 245 km. Above  $200\,\mathrm{km}$  altitude, the parametrisation is given by a constant function, i.e. one parameter. The altitude parametrisations of the corrections are illustrated in Figure 10. As for the temperature and atomic oxygen parametrisations, the first two B-spline coefficients are merged into two by zero curvature constraint at  $100\,\mathrm{km}$  altitude. The last three B-spline coefficients are fixed by  $C^2$ -continuity with the constant function at  $200\,\mathrm{km}$  altitude. Thus, the first and second order corrections each contain three free parameters. The corrections result in 12 (=3 × 2 × 2) additional parameters for the temperature and atomic oxygen retrieval.

**Figure 9.** LOS of three consecutive scans shown in different colors. To better see the individual LOS, only every fifth LOS is shown. The satellite orbit and direction are indicated by the black arc and the solid arrow. Gray circles indicate  $100\,\mathrm{km}$  altitude steps. The LOS of the scans can be seen to traverse a larger region which can be described by the angle  $\alpha$  with respect to the vector from the center of the earth and through the center of the three scans (grey solid line).

**Figure 10.** parametrisation of the first and second order corrections with altitude. The parametrisation contains three free parameters (two B-spline coefficients and one value for the constant function above 200 km altitude). Both the first and second order corrections for the temperature and the atomic oxygen concentration profiles utilises this parametrisation, however with their own set of parameters.

# 4.4 Solving the Inverse Problem

A retrieval is done by finding the parameters for the temperature and atomic oxygen parametrisations that best describe the measurements in a combined scan (three scans are combined for one retrieval). For this, a non-linear least squares approach is used to minimise the sum of squares of residuals between the simulated mission spectra and the corresponding simulations using the above described parametrisations (equation (7)). The retrieval of vertical wind profiles is outside the scope of this study. Instead, a free parameter for a Doppler shift of each of the simulated mission spectra is added to the retrieval. Adding such shifts, is equivalent to adding parameters for the averaged winds projected along the LOS.

For the non-linear least squares approach, we use the Gauss-Newton method (Rodgers, 2000). The Jacobian is numerically calculated in each iteration using the forward finite difference approximation. For the starting point in the iterative Gauss-Newton method, we use the set of parameters corresponding to the description of the NRLMSIS 2.1 2022-07-18, 00:00 UTC global average temperature profile shifted by +50 K and the global average atomic oxygen concentration profile multiplied by a factor of 0.5. The Doppler shifts/average LOS winds are all initialised at zero. No regularisation or smoothing is utilised.

In summary, we use 270 mission spectra (three scans) for the simultaneous retrieval of 15(=9+3+3) parameters describing the retrieved temperature profile, 15(=9+3+3) parameters describing the retrieved atomic oxygen concentration profile, and a parameter for each of the spectra describing the Doppler shifts due to LOS winds. One retrieval takes around eight minutes of computation time in Python with a 20 core CPU (Intel Xeon Gold 6242R). More information to the iterative Gauss-Newton method and an example of parameter convergence can be found in the supplementary materials.

#### 4.5 Remarks on Vertical resolution, Parametrisation, and Sampling

The individual line-of-sight (LOS) measurements traverse several hundred kilometers through the atmosphere. As a result, these measurements inherently offer poor vertical resolution. The vertical structure of the atomic oxygen and temperature are only reconstructed by the retrieval process, where the resolution is defined not by the measurements themselves, but by the vertical parametrisations. The vertical parametrisations are then limited by the field of view of the instrument defined by the diameter of the main mirror of the telescope (40 cm in this work).

The use of B-spline basis functions for the vertical parametrisation allows us to impose a controlled vertical resolution, chosen to accurately represent the variability seen in the NRLMSIS 2.1 model. The vertical resolution is approximately equal to the spacing of the B-spline knots. At lower altitudes, the knot spacing, and therefore the vertical resolution, is chosen to be finer. For example, at 100 km altitude, the B-spline knots for the temperature are 5.0 km apart and the FWHM of the B-spline basis function is 7.4 km. The difference in tangential heights between measurements in this area is 1 km, thus ensuring an oversampling of the B-spline basis function. At higher altitudes, the distances between the B-spline knots get larger (the B-spline basis functions get wider), and the tangential heights sampling become larger accordingly. The maximum tangential height sampling of 311 km was chosen to have one measurement at a tangential height above the altitudes of the extrapolating region in the atomic oxygen parametrisation which is at 300 km.

# 5 Retrieval Results

Retrievals have been performed for the series of scans corresponding to a full orbit of the satellite. The scans for a single retrieval have been combined with a sliding window, such that the first retrieval is obtained from all the simulated spectra in scan 1,2,3, and the second retrieval is obtained from all the spectra from scan 2,3,4, and so on. Thus, a vertical profile is obtained every 1250km in ground track distance. In total 31 retrievals have been performed. For all the retrievals, the parameters for the atomic oxygen and temperature parametrisations as well as the spectral shifts converge well and within approximately 15 iterations (see supplementary materials for an example). To illustrate the range of retrieval performance, we present detailed results for two examples: one with low residuals, showing a more accurate retrieval, and another with high residuals, reflecting a less accurate retrieval. Finally, the overall results are summarised for the full orbit.

The retrieved temperature and atomic oxygen profile from combining scan 1,2,3 can be seen in Figure 11. The retrieved profiles are shown together with the reference NRLMSIS 2.1 profiles as well as the start profiles used for the first iteration in the non-linear least squares approach. These start profiles have been used for all the retrievals. From  $100\,\mathrm{km}$  and until  $300\,\mathrm{km}$  altitude, the maximum relative deviations between the retrieved and the reference profiles can be seen to be around +2.5% and -3.5% for the temperature and atomic oxygen, respectively. This retrieval is an example of a more accurate retrieval.

In Figure 12, the retrieved temperature and atomic oxygen profile from combining scan 14,15,16 can be seen. The residuals are larger and within -3.5% and +30% for the temperature and atomic oxygen concentration, respectively. This retrieval is an example of a less accurate retrieval. In spite of the lower accuracy, the spectra corresponding to the retrieved profiles match the simulated spectra within the receiver noise. This can be seen from Figure 13 comparing spectra from scan 14,15,16 to the spectra corresponding to the retrieved profiles. For this retrieval, the  $\chi^2$ -value (sum of the squares of the residuals over all pixels of the spectra in the scan divided by the receiver noise) has been calculated to 31292. The probability of a larger  $\chi^2$ -value is then 87% indicating that the residuals are dominated by statistical fluctuations, i.e. receiver noise, and not parametrisation shortcomings.

Additionally, the fits yield frequency shifts of the line center. For the combined scan 14,15,16 the frequency shifts are in the range from  $(-195 \pm 32)$  KHz to  $(69 \pm 72)$  KHz for the 2.1 THz transition at tangential heights below  $140 \,\mathrm{km}$  altitude. These shifts are caused by the wind along the LOS and corresponds to speeds of  $(28 \pm 4)$  m/s and  $(-10 \pm 10)$  m/s, respectively.

In Figure 14, the retrieved temperature and the reference values are summarised for all scans. In both the retrieval and the reference profiles, an increase in temperature can be seen around locations close to the north and the south pole (see Figure 6 for global map of the scan centres). In general there is a good agreement between the retrieval and the reference temperatures.

In Figure 15, the retrieved atomic oxygen concentration can be seen on a logarithmic scale for all scans. There is a good agreement between the retrieval and the reference. A decrease in atomic oxygen concentration can be seen around the same positions as for the increase in temperature (in the polar regions). This anti-correlation of the temperature and atomic oxygen is present in the NRLMSIS 2.1 model and it is also captured in the retrieval. In Figure 16, the retrieved temperatures and atomic oxygen at a fixed altitude of 150 km can be seen for all the retrievals. From the figure, the anti correlation of the temperature and atomic oxygen can be seen better.

Figure 11. Retrieved profiles from the from scan 1,2,3 shown with the orange solid line. The shaded area around the retrieved profile indicates the uncertainties obtained from the least squares linearisation. The blue dashed line is the reference profile. The gray dotted line is the start profile used in the non-linear least squares solving. (a) The temperature profile. (b) Atomic oxygen concentration profile. The residuals between the retrieval and reference are also shown (black solid lines). For the temperature and the atomic oxygen concentration, the residuals are within +2.5% and -3.5%, respectively, from  $100 \, \mathrm{km}$  and until  $300 \, \mathrm{km}$  altitude.

In Figure 17, the relative deviations of retrieved and reference data as a function of altitude can be seen. The deviations have been averaged over all retrieved profiles. For both the temperature and the atomic oxygen, the average deviations can be seen to be relatively high for the lower altitudes. At  $100\,\mathrm{km}$  altitude the averaged deviations are within 2% and 14% for the temperature and the atomic oxygen, respectively. Above  $120\,\mathrm{km}$  altitude, the atomic oxygen concentration deviations are smaller and around 1%. At higher altitudes the temperature deviations increase again. This is due to the low atomic oxygen concentration and thus the low signal. The higher deviations for atomic oxygen at  $100\,\mathrm{km}$  altitude is due to a combination of high atomic oxygen concentrations and thus a strong self-absorption of the transitions and fewer measurements in the scans with LOS passing through such low altitudes. To access the impact of the simplified wind treatment in the retrievals (the approximation of LOS winds by Doppler shifts), the retrieval simulations were repeated for an atmosphere without winds. The resulting average deviations are also shown in Figure 17. The deviations are similar in both cases and the simplified wind treatment has little impact on the retrieval quality. Results from single scan retrievals without any corrections for spherical asymmetry can be found in the supplementary material.

Figure 12. Same as Figure 12, but for scan 14,15,16. The temperature and atomic oxygen residuals can be seen to be within -3.5% and +30%, respectively, for altitudes between  $100\,\mathrm{km}$  and  $300\,\mathrm{km}$ .

Figure 13. Examples of spectra from scan 14,15,16. In blue are the original simulated mission spectra. In orange are the spectra corresponding to the retrieved profiles and in grey are the spectra corresponding to the start profiles of the retrieval (see Figure 11). The tangential height, the time, the location of the satellite upon simulation, the integration time, and the simulated spectral noise (RMS) are annotated in the plots. The retrieved Doppler shifts ( $\Delta\nu$ ) and the  $1\sigma$  uncertainties are also annotated. (**a, b**) 2.1 THz and 4.7 THz spectra at tangential height 100 km. (**c, d**) 2.1 THz and 4.7 THz spectra at tangential height 121 km. (**e, f**) 2.1 THz and 4.7 THz spectra at tangential height 140 km. (**g, h**) 2.1 THz and 4.7 THz spectra at tangential height 249 km. The frequency axes correspond to the rest frequencies of the atomic oxygen transitions (see Table 3).

**Figure 14.** Heatmap showing the vertical temperature profiles for the different retrievals. (a) Retrieved vertical profiles at the combined scan centres. (b) Reference values at the scan centres. A good agreement can be seen between the reference and retrieved values.

Figure 15. Heatmap showing the vertical atomic oxygen concentration profiles for the different combined scans. (a) Retrieved vertical profiles at the combined scan centers. (b) Reference values at the scan centers. Values below  $10^{15} \,\mathrm{m}^{-3}$  are shown with the same color/nuance. A good agreement can be seen between the reference and retrieved values. Note the logarithmic color axis.

Figure 16. The retrieved temperature and atomic oxygen at an altitude of 150 km versus the scans shown together with the reference/N-RLMSIS values. The crosses indicate the reference values (blue/dark toned: temperature and orange/light toned: atomic oxygen). The vertical bars indicate the retrievals as one sigma confidence intervals (blue/dark toned: temperature and orange/light toned: atomic oxygen).

**Figure 17.** Average deviations for all retrieved temperature and atomic oxygen vertical profiles. Temperature deviations are shown in blue/dark tones, and atomic oxygen deviations in orange/light tones. Dashed curves show results for an atmosphere without winds. The similarity between cases indicates that the simplified wind treatment has little impact on retrieval quality.

# 6 Conclusion and Outlook

Using the NRLMSIS 2.1 and the HWM14 atmosphere models, spectra of the  $2.1\,\mathrm{THz}$  and  $4.7\,\mathrm{THz}$  atomic oxygen transitions as measured by a limb sounding satellite were simulated. Using realistic THz receiver noise temperatures, atomic oxygen concentration were retrieved from  $100\,\mathrm{km}$  to  $110\,\mathrm{km}$  within  $\pm 15\%$  average deviations and from  $110\,\mathrm{km}$  to  $300\,\mathrm{km}$  within  $\pm 3\%$  average deviations. Temperatures were retrieved within  $\pm 3\%$  average deviations from  $100\,\mathrm{km}$  to  $175\,\mathrm{km}$ . Although the retrieval of winds was not the scope of this study, average winds along the LOS were retrieved and appear to be a good first approximation for capturing wind effects on the observed spectra. Due to the high flexibility of the retrieval scheme, the majority of the retrieval uncertainties are concluded to stem from receiver noise and not parametrisation inefficiencies. This is further supported by the agreement of the simulated spectra and the spectra corresponding to the retrieved profiles.

Noise temperatures of 11,000 K and 25,000 K were assumed for the 2.1 THz and 4.7 THz channels, respectively. If receivers could be made with lower noise, the atomic oxygen and temperature could be retrieved with smaller uncertainties or the retrieval could be extended to lower altitudes of the atmosphere. For an example the retrieval could start at 80 km instead of 100 km. However, due to the high oxygen concentration from 80 km to 100 km altitude (as expected from the NRLMSIS 2.1 model) and thereby the strong self-absorption in measurements at tangential heights between 80 km and 100 km altitude, the retrieval uncertainties would likely only be improved by a smaller fraction. By the same argument, uncertainties in pointing of the satellite as well as intensity calibrations would also be more critical around such altitudes. Pointing and calibration uncertainties were not considered in this study.

If a smaller altitude range of the atmosphere is to be investigated, such as the region between 100 km and 150 km, it could potentially increase precision or allow for smaller ground track sampling. However, the tangential height measurements in the smaller region remain influenced by the oxygen and temperature conditions at higher altitudes. Therefore, it is not immediately clear whether a reduced altitude range would enhance precision for the lower altitudes. This depends on how accurately the oxygen and temperature profiles can be extrapolated above 150 km where no tangential height measurements would be and how the uncertainties of the atomic oxygen and temperatures above 150 km propagate to the altitudes below. To evaluate this, one would therefore have to redefine the mission scenario with tangential height measurements focused specifically on the  $100 \, \mathrm{km}$  to  $150 \, \mathrm{km}$  altitude range and redo the retrievals accordingly.

The results of this study highlight the feasibility of accurate temperature and atomic oxygen retrieval at altitudes above of  $100\,\mathrm{km}$ . This supports the development of future space missions, such as the Keystone, exploring the photochemistry and dynamics of the MLT region.

Code availability. Code is not available.

Data availability. Simulated mission spectra are available from the corresponding author upon request.

# **Appendix A: Convergence During Retrieval**

In the retrieval process, the parameters describing temperature, atomic oxygen concentration, and spectral shifts are estimated using the iterative Gauss-Newton method. In this approach, the forward problem, i.e. simulating the spectra used for the given retrieval, is linearised around the current estimates of these parameters. We do that using the forward difference method. The resulting linearisation allows the parameters to be updated through a linear least-squares solution yielding improved estimates. For the least-squares solution, we use the function  $lsq\_linear$  from the scipy library. To avoid oscillations of the parameter estimates around the minimum, we update the parameters as an weighted average between the newly found least-squares solution and the parameters in the previous iteration such that

315 
$$x_{i+1} = 0.6 \cdot x_{i+1}^{lsq\_linear} + 0.4 \cdot x_i$$
, (A1)

where  $x_{i+1}$  is the vector containing the improved parameter estimates,  $x_i$  is the vector containing the parameter estimates in the previous iteration, and  $x_{i+1}^{lsq\_linear}$  is the parameter estimates from the least squares solution. We stop the iterative process when the temperature and atomic oxygen parameters change less than  $10^{-3}$ , the sum of all the shifts less than  $10 \, \mathrm{kHz}$ , and the residuals, relative to the residuals in the previous step, change less than  $10^{-6}$ . Figure A1 shows the parameter values during the retrieval from combining scan 14,15,16, where convergence occurs within 15 iterations. The convergence behavior for the remaining retrievals is similar and is therefore not shown.

# **Appendix B: Single Scan Retrievals Versus Combined Scan Retrievals**

To assess the significance of the second-order correction for spherical asymmetry, retrievals using single scans without any correction were also performed. Figure B1 shows the resulting mean deviations in temperature and atomic oxygen concentrations versus altitude, comparing them to the deviations from retrievals using combined scans with the second-order correction applied (Figure 17). The deviations from the single scans are significantly larger. For an example, at an altitude of 110 km, the single-scan deviations are approximately five times larger. This exceeds the expected factor of square-root three, which would result solely from the increased signal provided by combining three scans. These results demonstrate that applying spherical asymmetry corrections is essential for achieving accurate retrievals for the proposed mission scenario.

Figure A1. Parameter values at each iteration in the Gauss Newton method for the retrieval at the 14th combined scan (scan 14,15,16). (a, b, c) Parameters of the temperature parametrisation. (d, e, f) Parameters of the atomic oxygen parametrisation. (g) The parameters for the shifts. The parameters have been scaled/shifted upon retrieval for obtaining values of around  $\pm 1$ . For  $T_{sym}$  and  $\rho_{sym}$ , parameter numbers from 1 to 6 and 1 to 7, respectively, corresponds to the B-spline coefficients, in increasing altitude order. For an example, the B-spline coefficient for  $T_{sym}$  are scaled by  $10^{-3}$ . The shifts are scaled by  $10^{-6}$  (scaled to MHz). Due to the many parameters, the shift's parameters are not numbered.

**Figure B1.** Average deviations for all the retrieved temperature and atomic oxygen concentration vertical profiles. Dashed grey line showing the average deviations from single scan retrievals and black solid line showing the average deviations from combined scans with second order correction for spherical asymmetry. (a) Temperature deviations. (b) Atomic oxygen concentration deviations.

| 330 | Author contributions. HWH, MW, and PBH all contributed to the concept and idea of the paper. PBH carried out simulations and data |
|-----|-----------------------------------------------------------------------------------------------------------------------------------|
|     | analyses with support from MW and HWH. PBH prepared the manuscript with contributions from MW and HWH.                            |

Competing interests. The authors declare that they have no conflict of interest.

Acknowledgements. We gratefully acknowledge funding by the Deutsche Forschungsgemeinschaft (DFG, German Research Foundation, project number 502949516).

365

- ESA selects four new Earth Explorer mission ideas, https://www.esa.int/Applications/Observing the Earth/FutureEO/Preparing for tomorrow/ESA selects four new Earth Explorer mission ideas, [Online; accessed 20-August-2024], 2024.
- Bates, D. R. and Massey, H. S. W.: Some problems concerning the terrestrial atmosphere above about the 100 km level. Proceedings of the Royal Society of London. Series A. Mathematical and Physical Sciences, 253, 451-462, https://doi.org/10.1098/rspa.1959.0207, 1959.
- 340 de Boor, C.: A Practical Guide to Splines, in: Applied Mathematical Sciences, https://api.semanticscholar.org/CorpusID:122101452, 1978.
  - Drob, D. P., Emmert, J. T., Meriwether, J. W., Makela, J. J., Doornbos, E., Conde, M., Hernandez, G., Noto, J., Zawdie, K. A., McDonald, S. E., Huba, J. D., and Klenzing, J. H.: An update to the Horizontal Wind Model (HWM): The quiet time thermosphere, Earth and Space Science, 2, 301–319, https://doi.org/https://doi.org/10.1002/2014EA000089, 2015.
- Emmert, J. T., Jones Jr. M., Siskind, D. E., Drob, D. P., Picone, J. M., Stevens, M. H., Bailey, S. M., Bender, S., Bernath, P. F., Funke, 345 B., Hervig, M. E., and Pérot, K.: NRLMSIS 2.1: An Empirical Model of Nitric Oxide Incorporated Into MSIS, Journal of Geophysical Research: Space Physics, 127, e2022JA030896, https://doi.org/https://doi.org/10.1029/2022JA030896, e2022JA030896 2022JA030896, 2022.
  - Gaidis, M., Pickett, H., Smith, C., Martin, S., Smith, R., and Siegel, P.: A 2.5-THz receiver front end for spaceborne applications, IEEE Transactions on Microwave Theory and Techniques, 48, 733–739, https://doi.org/10.1109/22.841966, 2000.
- Grossmann, K. U., Kaufmann, M., and Gerstner, E.: A global measurement of lower thermosphere atomic oxygen densities, Geophysical Research Letters, 27, 1387–1390, https://doi.org/https://doi.org/10.1029/2000GL003761, 2000.
  - Jacchia, L. G.: Static diffusion models of the upper atmosphere with empirical temperature profiles, Smithsonian Contributions to Astrophysics, 8, 213–257, https://doi.org/10.5479/si.00810231.8-9.213, 1965.
- Kramida, A., Ralchenko, Y., Reader, J., and NIST Atomic Spectra Database Team: NIST Atomic Spectra Database (version 5.10), 355 https://doi.org/https://doi.org/10.18434/T4W30F, 2022.
  - Livesey, N. J. and Read, W. G.: Direct retrieval of line-of-sight atmospheric structure from limb sounding observations, Geophysical Research Letters, 27, 891–894, https://doi.org/https://doi.org/10.1029/1999GL010964, 2000.
  - Mlynczak, M. G. and Solomon, S.: A detailed evaluation of the heating efficiency in the middle atmosphere, Journal of Geophysical Research: Atmospheres, 98, 10517-10541, https://doi.org/https://doi.org/10.1029/93JD00315, 1993.
- 360 Mlynczak, M. G., Hunt, L. A., Russell III, J. M., and Marshall, B. T.: Updated SABER Night Atomic Oxygen and Implications for SABER Ozone and Atomic Hydrogen, Geophysical Research Letters, 45, 5735-5741, https://doi.org/https://doi.org/10.1029/2018GL077377, 2018.
  - Offermann, D., Grossmann, K.-U., Barthol, P., Knieling, P., Riese, M., and Trant, R.: Cryogenic Infrared Spectrometers and Telescopes for the Atmosphere (CRISTA) experiment and middle atmosphere variability, Journal of Geophysical Research: Atmospheres, 104, 16311– 16 325, https://doi.org/https://doi.org/10.1029/1998JD100047, 1999.
  - Richter, H., Buchbender, C., Güsten, R., Higgins, R., Klein, B., Stutzki, J., Wiesemeyer, H., and Hübers, H.-W.: Direct measurements of atomic oxygen in the mesosphere and lower thermosphere using terahertz heterodyne spectroscopy, Communications Earth & Environment, 2, 19, https://doi.org/10.1038/s43247-020-00084-5, 2021.
- Riese, M., Offermann, D., and Brasseur, G.: Energy released by recombination of atomic oxygen and related species at mesopause heights, 370 Journal of Geophysical Research: Atmospheres, 99, 14585-14593, https://doi.org//https://doi.org/10.1029/94JD00356, 1994.
  - Rodgers, C. D.: Inverse Methods for Atmospheric Sounding, WORLD SCIENTIFIC, https://doi.org/10.1142/3171, 2000.

- Sharma, R., Zachor, A., and Yap, B.: Retrieval of atomic oxygen and temperature in the thermosphere. II—feasibility of an experiment based on limb emission in the OI lines, Planetary and Space Science, 38, 221–230, https://doi.org/https://doi.org/10.1016/0032-0633(90)90086-6, 1990.
- Sharma, R., Zygelman, B., von Esse, F., and Dalgarno, A.: On the relationship between the population of the fine structure levels of the ground electronic state of atomic oxygen and the translational temperature, Geophysical Research Letters, 21, 1731–1734, https://doi.org/https://doi.org/10.1029/94GL01078, 1994.

380

395

- Sheese, P. E., McDade, I. C., Gattinger, R. L., and Llewellyn, E. J.: Atomic oxygen densities retrieved from Optical Spectrograph and Infrared Imaging System observations of O2 A-band airglow emission in the mesosphere and lower thermosphere, Journal of Geophysical Research: Atmospheres, 116, https://doi.org/10.1029/2010JD014640, 2011.
- Siles, J. V., Maestrini, A. E., Lee, C., Lin, R., and Mehdi, I.: First Demonstration of an All-Solid-State Room Temperature 2-THz Front End Viable for Space Applications, IEEE Transactions on Terahertz Science and Technology, 14, 607–612, https://doi.org/10.1109/TTHZ.2024.3430013, 2024.
- Wienold, M., Semenov, A. D., Dietz, E., Frohmann, S., Dern, P., Lü, X., Schrottke, L., Biermann, K., Klein, B., and Hübers, H.-W.: OSAS B: A Balloon-Borne Terahertz Spectrometer for Atomic Oxygen in the Upper Atmosphere, IEEE Transactions on Terahertz Science and Technology, 14, 327–335, https://doi.org/10.1109/TTHZ.2024.3363135, 2024.
  - Wiesemeyer, H., Güsten, R., Aladro, R., Klein, B., Hübers, H.-W., Richter, H., Graf, U. U., Justen, M., Okada, Y., and Stutzki, J.: First detection of the atomic <sup>18</sup>O isotope in the mesosphere and lower thermosphere of Earth, Phys. Rev. Res., 5, 013 072, https://doi.org/10.1103/PhysRevResearch.5.013072, 2023.
- Wu, D. L., Yee, J.-H., Schlecht, E., Mehdi, I., Siles, J., and Drouin, B. J.: THz limb sounder (TLS) for lower thermospheric wind, oxygen density, and temperature, Journal of Geophysical Research: Space Physics, 121, 7301–7315, https://doi.org/https://doi.org/10.1002/2015JA022314, 2016.
  - Zachor, A. and Sharma, R.: Retrieval of atomic oxygen and temperature in the thermosphere. I Feasibility of an experiment based on the spectrally resolved 147 μm limb emission, Planetary and Space Science, 37, 1333–1346, https://doi.org/https://doi.org/10.1016/0032-0633(89)90105-0. 1989.
  - Zink, L. R., Evenson, K. M., Matsushima, F., Nelis, T., and Robinson, R. L.: Atomic oxygen fine-structure splittings with tunable far-infrared spectroscopy, Astrophysical Journal; (United States), 371, https://doi.org/10.1086/186008, 1991.