# Peer review of "Feasibility of a space-borne terahertz heterodyne spectrometer for atomic oxygen and temperature in the mesosphere and lower thermosphere"

_EGUsphere, 2024_

## Referee Comment (RC1)

**Review of "Feasibility of a space-borne terahertz heterodyne spectrometer for atomic oxygen and temperature in the mesosphere and lower thermosphere" by Peder Bagge Hansen et al.**

**Overview**

This paper is a feasibility study for retrieving atomic oxygen concentration and temperature from 2.1 THz and 4.7 THz atomic oxygen emission spectra measured by a limb-sounding heterodyne spectrometer from a satellite in polar orbit. The main idea of the study is generating simulated measurement spectra (based on temperature and atomic oxygen density from NRLMSIS 2.1 and winds from the HWM14 model) and retrieving temperature and atomic oxygen profiles from them. Atmosphere is not treated as spherically symmetric by the retrieval: three consecutive atmospheric scans are considered at a time, thus improving the retrieval of horizontal structures. Feasibility of retrieving line-of-sight winds from these spectra is also briefly discussed.

In general the paper is well-written, well-organised, methods and results are generally very clearly presented. The authors make a convincing case that the kind of retrieval they demonstrate here is feasible. My main points of criticism (see General/Major comments for more detail) are that the vertical resolution limits for this retrieval were not really explored (the retrieved data has much coarser vertical sampling than the simulated atmosphere scans), and that authors show very little data on retrieved winds. I would have also liked to see a few more comments on why some of the methods were chosen.

From the technical perspective, the quality of the manuscript is excellent. I have found very few typos, and made only a couple of small language and terminology suggestions.

**General/major comments**

1. **Vertical resolution.** The retrieval setup described in the manuscript seems to be aimed at delivering high vertical resolution: the vertical sampling of the measurements in a scan is 1 km of tangent altitude at the bottom of the altitude range, and the instrument has "1.0km field of view (FWHM) at the 100km tangential point for the 2.1THz receiver and 0.45km for the 4.7THz receiver". However, the authors chose to represent their retrieved temperature and atomic oxygen concentration profiles with a set of B-splines, which have the knots spaced 5 km apart at the lower end of the altitude range, closely following the way atmospheric quantities are represented in the NRLMSIS model used as a source for the simulated data. I believe this representation (although seemingly perfectly valid mathematically) unnecessarily limits the performance of the retrieval. In particular, the vertical sampling of the retrieved quantities ends up being much coarser than the vertical sampling of the simulated measurements. Therefore, there is no way to tell if the proposed measurement scheme can deliver the high resolution at the lower altitudes it seems to be designed to achieve.

   I understand that the authors want their simulated measurements to be based on realistic data, and it can be difficult to obtain upper atmosphere data with highly realistic small vertical scale structures. However, one could simply resample NRLMSIS data on a denser vertical grid (or just use B-spline representation with more splines) and add some arbitrary small vertical scale structures to the reference temperature and atomic oxygen profiles just to see if they can be faithfully retrieved. I believe that a vertical resolution test with not-so-realistic data would be better than effectively not exploring the vertical resolution limits at all.

   This is particularly relevant in the light of some of the results presented in the manuscript. Figures 12 and 17 clearly show that 100 km to 125 km altitude region seems to be the most

problematic, the retrieved profiles fluctuate around the "true" values much more than in any other altitude range. Furthermore, the authors say that "the majority of the retrieval uncertainties are concluded to stem from receiver noise and not parametrisation inefficiencies". If that is the case, and retrievals in this altitude range are inherently less stable even with 5 km vertical sampling (distance between B-spline knots), then is it really worthwhile to use 1 km vertical sampling for the measurements? Perhaps it would be better to reduce the scanning time and thus improve horizontal resolution (make ground track sampling denser) instead?

More generally, the authors seem to have taken great care to keep number of retrieved parameters to absolute minimum. While it is, generally speaking, a good practice, the ratio between the amount of retrieved parameters and the number of measurements is quite low in this study. This is not a problem in itself, but if the number of retrieved variables was kept so low because introducing any more lead to serious problems, maybe the dense altitude sampling used here is not bringing the expected benefits?

It would be good if the authors could at least add some comments explaining their choice of vertical sampling and the main limiting factors on the vertical resolution of the proposed instrument.

2. **Wind retrievals**

   (a) The authors state that retrieval of wind profiles is outside the scope of the study, and that one free parameter for Doppler shift was added to each simulated spectrum. I would argue, that since most of the radiance observed along each LOS comes from the vicinity of the tangent point, retrieving one LOS wind value per measurement is a rough approximation of a retrieval of a vertical wind profile (wind projection to LOS direction), just with a highly simplified wind treatment in radiative transfer calculations. It would therefore be good to show the retrieved wind values as a function of tangent point altitude. Even if the authors do not claim that they have retrieved the winds well, it is important to show the results in detail, since any problems with wind retrieval could have had a significant effect on the retrieval of other quantities. In other words, since winds were included into generation of simulated measurements, I think they are within the scope of the study, even if the authors only claim to have achieved good results for temperature and atomic oxygen retrievals.

   (b) As far as I understood from the general measurement setup, 2.1THz and 4.7THz measurements would be taken simultaneously for a given tangent point altitude within a given scan, and then modelled using the same LOS. However, the Doppler shifts for winds in these two measurements are retrieved as independent from one another. Is that correct? If yes, would it not be more logical to retrieve one wind value per LOS, from which the corresponding Doppler shifts could be derived? This would reflect the fact that both channels of the instrument should hopefully see the same wind?

3. **Non-linearity of inversion.** It is generally considered very desirable to keep inverse modelling problems as linear as possible. This both allows one to use many standard results of inverse modelling theory (Rodgers (2000) that the authors cite contains a lot of them), reduce computational costs with highly efficient linear algebra libraries and generally make retrievals more stable. Radiative transfer equations, sadly, are often non-linear, but most authors try not to introduce additional non-linearity beyond the forward model. Therefore, I find the equation (7) of this manuscript quite odd. It seems to me that one could have simply used additive correction terms instead of the multiplicative ones. I have briefly skimmed Livesey and Reed (2000), that the authors cite as an inspiration for their spherical asymmetry correction. That work does indeed introduce the idea of handling multiple adjacent vertical profiles together for improved retrieval performance, but standard linear algebra techniques seem to be used there. Could the authors

comment on why this approach was chosen?

**Minor/specific comments**

1. Lines 21-22: I am not quite sure what the authors mean by "[...] systematically analyses concentric atmospheric layers". I would say that the main intrinsic advantage of limb observation geometry is high vertical resolution. Onion peeling, on the other hand, is just one of the methods (and perhaps the simplest one), to do a limb retrieval. In my opinion, the fact that onion peeling retrieves the parameters of the outer layers first, and uses them as the "truth" to retrieve the inner layers is more of a flaw than an advantage.

2. Line 43: I would replace "non-spherical symmetric atmosphere" with "non-spherically symmetric atmosphere". More generally, the authors mention this lack of spherical symmetry quite a few times, but the first really clear explanation of what they mean is given in line 176: " [...] variations in the vertical profiles across different longitudes and latitudes, i.e. the spherical asymmetry of the atmosphere [...]". I think this could be made clearer by either giving this explanation right from the beginning (i.e. line 43), or using some other term, like "horizontal structure of the atmosphere" instead of "spherical asymmetry". After line 43, I was left wondering whether the authors mean the horizontal structure, or do they mean that the oblateness of the Earth was accounted for in their calculations.

**Minor typos and suggestions**

Here is the list of minor suggestions and typos that I have noticed. I do not expect point-by-point answers for these.

1. Line 59: Replace "5.2MHz (FHWM)" with "5.2MHz (FWHM)".

2. Line 184: "Without introduction of $\alpha$, the retrieved profiles would correspond to some kind of averages over the regions traversed by the LOS." It seems to me that the benefit of correcting for spherical asymmetry in general is described here, and not an advantage of the particular parametrisation of the horizontal structure (i.e. the $\alpha$ angle). It would be good to make this clear.

3. Line 238: "This retrieval resembles the example of a more accurate retrieval." Do the authors simply mean that this retrieval *is* an example of a more accurate retrieval? If it only "resembles the example", then where is that other example shown or described? The same applies for the similar statement in line 241.

4. Penultimate line of the Figure 13 caption: replace "corresponds" with "correspond".

---

## Author Comment (AC1)

We thank the reviewer for the thoughtful and thorough comments. They have sparked a good discussion. We highly appreciate the time that was put into this.

Please find our replies below in green.

General/major comments:

1. **Vertical resolution.** The retrieval setup described in the manuscript seems to be aimed at delivering high vertical resolution: the vertical sampling of the measurements in a scan is 1 km of tangent altitude at the bottom of the altitude range, and the instrument has "1.0km field of view (FWHM) at the 100km tangential point for the 2.1THz receiver and 0.45km for the 4.7THz receiver". However, the authors chose to represent their retrieved temperature and atomic oxygen concentration profiles with a set of B-splines, which have the knots spaced 5 km apart at the lower end of the altitude range, closely following the way atmospheric quantities are represented in the NRLMSIS model used as a source for the simulated data. I believe this representation (although seemingly perfectly valid mathematically) unnecessarily limits the performance of the retrieval. In particular, the vertical sampling of the retrieved quantities ends up being much coarser than the vertical sampling of the simulated measurements. Therefore, there is no way to tell if the proposed measurement scheme can deliver the high resolution at the lower altitudes it seems to be designed to achieve.

   We choose a narrow field of view ("1.0km field of view (FWHM) at the 100km tangential point for the 2.1THz receiver and 0.45km for the 4.7THz receiver") in order to approximate the field-of-view (FOV) geometry by a single LOS. This makes the calculations involved in the retrieval faster since the field of view does not need to be approximated by several line-of-sight calculations. This would also be beneficial for a real mission where much more data is to be analyzed. Furthermore, we didn't want the FOV to be the imitating factor in the vertical resolution.

   It is correct that our tangential height sampling is finer than the resolution of the retrieved values. By using a cruder tangential height sampling, we could have a better signal-to-noise ratio in each of the measurements. But on the other hand, we would have fewer measurements and the total amount of information/signal-to-noise would be approximately the same. Thus, we don't believe that our tangential height sampling limits the performance as long as we are oversampling the retrieved quantities.

   I understand that the authors want their simulated measurements to be based on realistic data, and it can be difficult to obtain upper atmosphere data with highly realistic small vertical scale structures. However, one could simply resample NRLMSIS data on a denser vertical grid (or just use B-spline representation with more splines) and add some arbitrary small vertical scale structures to the reference temperature and atomic oxygen profiles just to see if they can be faithfully retrieved. I believe that a vertical resolution test with not-so-realistic data would be better than effectively not exploring the vertical resolution limits at all.

   This is a really good idea and probably the best way of exploring the vertical resolution of the retrieval in depth. But, this would also require a large amount of extra work and we think that this would rather be the topic of a separate study than a nice addition to this one.
   For a small discussion of the vertical resolution see our comments later in this document concerning the section "Remarks on Parametrisation and Tangential Height Sampling" in the preprint.

This is particularly relevant in the light of some of the results presented in the manuscript. Figures 12 and 17 clearly show that 100 km to 125 km altitude region seems to be the most problematic, the retrieved profiles fluctuate around the "true" values much more than in any other altitude range. Furthermore, the authors say that "the majority of the retrieval uncertainties are concluded to stem from receiver noise and not parametrisation inefficiencies". If that is the case, and retrievals in this altitude range are inherently less stable even with 5 km vertical sampling (distance between B-spline knots), then is it really worthwhile to use 1 km vertical sampling for the measurements? Perhaps it would be better to reduce the scanning time and thus improve horizontal resolution (make ground track sampling denser) instead?

Reducing the vertical sampling of the measurements, without increasing the integration time accordingly, would result in larger retrieval uncertainties and thereby more fluctuations around the "true" values. So, we don't think it would be better to improve the horizontal sampling.

On the other hand one could argue that the uncertainties around 100 km altitude are so high that they don't justify the high resolution of the parametrisation. Or put differently, one could argue that we are overfitting the atomic oxygen concentration at densities around 100 km altitude. But we think it is more educational to keep the high vertical resolution since otherwise our retrieval uncertainties would not only be dominated by receiver noise but also by the parametrisation.

More generally, the authors seem to have taken great care to keep number of retrieved parameters to absolute minimum. While it is, generally speaking, a good practice, the ratio between the amount of retrieved parameters and the number of measurements is quite low in this study. This is not a problem in itself, but if the number of retrieved variables was kept so low because introducing any more lead to serious problems, maybe the dense altitude sampling used here is not bringing the expected benefits?

Yes, we have indeed taken great care to keep the number of retrieved parameters to an absolute minimum. Increasing the number of parameters in the retrieval naturally results in larger retrieval uncertainties. This can be seen in Figure 1 showing the retrieved vertical profiles and deviations from combined scan 1,2,3 (Figure 11 in preprint) using approximately twice the number of B-splines (~2.5 km knot separation at 100 km altitude) and 2x5 parameters for the horizonal correction terms instead of 2x3. The corresponding parameter convergence is shown in Figure 2. From the figure, it can be seen that increasing the number of retrieval parameters doesn't lead to any serious problems such as fail of convergence.

[Figure]

*Figure 1. Retrieval results for scan 1,2,3 with a more flexible parametrization containing approximately twice the number of parameters compared to the parametrisation used in the preprint.*

[Figure]

*Figure 2. Parameter convergence for the retrieval shown in Figure 1. The retrieval is for a parametrization of approximately twice of that in the preprint. The parameters for the atomic oxygen and temperature can be seen to converge well.*

It would be good if the authors could at least add some comments explaining their choice of vertical sampling and the main limiting factors on the vertical resolution of the proposed instrument.

We have rewritten section 4.5 "Remarks on Parametrisation and Tangential Height Sampling" to better address the vertical resolution. The section now reads:

"**4.5 Remarks on Vertical resolution, Parametrisation and Sampling**

The individual line-of-sight (LOS) measurements traverse several hundred kilometers through the atmosphere. As a result, these measurements inherently offer poor vertical resolution. The vertical structure of the atomic oxygen and temperature are only reconstructed by the retrieval process, where the resolution is defined not by the measurements themselves, but by the vertical parametrisations. The vertical parametrisations are then limited by the field of view of the instrument defined by the diameter of the main mirror of the telescope (40 cm in this work).

The use of B-spline basis functions for the vertical parametrisation allows us to impose a controlled vertical resolution, chosen to accurately represent the variability seen in the NRLMSIS 2.1 model. The vertical resolution is approximately equal to the spacing of the B-spline knots. At lower altitudes, the knot spacing, and therefore the vertical resolution, is chosen to be finer. For example, at 100km altitude, the B-spline knots for the temperature are 5.0 km apart and the FWHM of the B-spline basis function is 7.4 km. The difference in tangential heights between measurements in this area is 1 km, thus ensuring an oversampling of the B-spline basis function. At higher altitudes, the distances between the B-spline knots get larger (the B-spline basis functions get wider), and the tangential heights sampling become larger accordingly. The maximum tangential height sampling of 311km was chosen to have one measurement at a tangential height above the altitudes of the extrapolating region in the atomic oxygen parametrisation which is at 300km."

2. Wind retrievals
   a. The authors state that retrieval of wind profiles is outside the scope of the study, and that one free parameter for Doppler shift was added to each simulated spectrum. I would argue, that since most of the radiance observed along each LOS comes from the vicinity of the tangent point, retrieving one LOS wind value per measurement is a rough approximation of a retrieval of a vertical wind profile (wind projection to LOS direction), just with a highly simplified wind treatment in radiative transfer calculations. It would therefore be good to show the retrieved wind values as a function of tangent point altitude. Even if the authors do not claim that they have retrieved the winds well, it is important to show the results in detail, since any problems with wind retrieval could have had a significant effect on the retrieval of other quantities. In other words, since winds were included into generation of simulated measurements, I think they are within the scope of the study, even if the authors only claim to have achieved good results for temperature and atomic oxygen retrievals.

   Below is a figure of the temperature, atomic oxygen, and line-of-sight (LOS) winds (from MSIS and HWM) for a measurement at 100 km tangential height for a satellite at zero longitude and latitude and 500 km altitude looking straight north. As seen from the figure, the wind along the LOS changes direction along the horizontal span of the LOS. For an example: the first time the LOS crosses 300 km altitude the wind is

around 0 m/s. The second time it crosses 300 km altitude the wind is around -50 m/s. On the contrary, the temperature and atomic oxygen change only slightly with the horizontal position. The reference winds at the tangent points are therefore poorly defined without any horizontal variable. This is why we try not to get into the wind retrieval discussion.

[Figure]

*Figure 3.. (a,b,c) Sample LOS temperature, atomic oxygen, LOS winds for a measurement at 100 km tantgential height. (d) Visualization of the LOS also indicating the horizontal direction and the start of the LOS by the circle and the "end" of the LOS by the arrow.*

We have performed retrieval simulations for an atmosphere with zero winds, i.e. we ignore winds when simulating mission data and when performing the retrieval. The mean atomic oxygen deviations for the full orbit (31 retrievals) are shown in Figure 3 together with the deviations when considering winds and compensating with a spectral shift (what is shown in the preprint). It can be seen that the deviations are within a few percentages from each other and we conclude that the simplified wind treatment does not result in any significant degradation of atomic oxygen retrieval.

[Figure]

*Figure 4. Atomic oxygen retrieval deviations for an atmosphere with and without winds.*

As requested, the retrieved wind values are shown as a function of tangent point altitude in Figure 5. Although interesting, we prefer to keep the wind retrieval discussion for a future detailed model. At this stage our analysis shows that the uncertainty of the temperature and oxygen retrieval is sufficiently low even if the winds are not considered in detail.

[Figure]

*Figure 5. Examples of retrieved Doppler shifts and the LOS winds from the HWM (i.e. the reference values) at the tangential point altitudes. Left: for the scan 1,2,3 retrieval. Right: for the scan 6,7,8 retrieval. For 2THz, 4THz, and the reference values there are three data points per tangential height because that the retrievals are done for three scans.*

b. As far as I understood from the general measurement setup, 2.1THz and 4.7THz measurements would be taken simultaneously for a given tangent point altitude within a given scan, and then modelled using the same LOS. However, the Doppler shifts for winds in these two measurements are retrieved as independent from one another. Is that correct? If yes, would it not be more logical to retrieve one wind value per LOS, from which the corresponding Doppler shifts could be derived? This would reflect the fact that both channels of the instrument should hopefully see the same wind?

Yes, that is correctly understood and we agree that the 2.1 THz and 4.7 THz pairs of measurements should have the same Doppler shifts. However, the Doppler shifts determined from the measurements at 2.1 THz and 4.7 THz are not exactly the same for the following reason: The 4.7 THz measurements are stronger saturated than the 2.1 THz measurements at tangential heights around 100 km. Consequently, the signal contribution from the high-altitude layers is slightly larger for the 4.7 THz measurements than for the 2.1 THz measurements. This can be inferred from Figure 6 where the 4.7 THz solved Doppler shifts are seen to be different from the 2.1 THz solved Doppler shifts. The 4.7 THz Doppler shifts are more biased towards the winds at the higher altitudes.

[Figure]

*Figure 6. The same data as in Figure 5 (left) shown in a smaller tangential height range. Although the 2.1 THz and 4.7THz measurements have the same LOS geometry, the retrieved doppler shifts are different indicating a different influence on the 2.1 THz and 4.7 THz spectra from different atmospheric altitudes/layers.*

3. **Non-linearity of inversion.** It is generally considered very desirable to keep inverse modelling problems as linear as possible. This both allows one to use many standard results of inverse modelling theory (Rodgers (2000) that the authors cite contains a lot of them), reduce computational costs with highly efficient linear algebra libraries and generally make retrievals more stable. Radiative transfer equations, sadly, are often non-linear, but most authors try not to introduce additional non-linearity beyond the forward model. Therefore, I find the equation (7) of this manuscript quite odd. It seems to me that one could have simply used additive correction terms instead of the multiplicative ones. I have briefly skimmed Livesey and Reed (2000), that the authors cite as an inspiration for their spherical asymmetry correction. That work does indeed introduce the idea of handling multiple adjacent vertical profiles together for improved retrieval performance, but standard linear algebra techniques seem to be used there. Could the authors comment on why this approach was chosen?

In Livesey and Reed (2000) they describe how to combine "x" scans for the simultaneous retrieval and interpolation of "x" vertical profiles. With the approach of Livesey and Reed

(2000) we would have combined three scans to retrieve three vertical profiles resulting in 9 + 9 + 9 parameters (three full vertical profiles). In our approach, we combine three scans to retrieve 9 + 3 + 3 parameters. 9 parameters for the vertical profile and 3 + 3 parameters for the horizontal correction (alpha direction). Thus, we increase the information by a factor of three (combining three scans) and only increase the parameters by a factor of 1.67. We do this because the horizontal variation of the vertical profiles is expected to be much smaller than the vertical variation.

Why don't we use an additive correction? We have found that a relative correction can describe the horizontal variation with fewer parameters than an additive correction term. This is shown in Figure 7.

It is correct that the type of correction we use is non-linear, however the inversion is already non-linear and thus there's no win in terms of computation time by using an additive correction. Also, we observe no problems with the convergence of the horizontal correction parameters.

[Figure]

[Figure]

Figure 7. (a) Parametrisation deviations of the NRLMSIS 2.1 atomic oxygen concentration along the lines-of-sight illustrated in (b). Orange curves are the parametrisation deviations using an additive type of horizontal correction. Blue curves are the parametrisation deviations using a relative horizontal correction. The parametrisation deviations can be seen to be lower when using a relative horizontal correction term.

*It is important to stress that this is no retrieval, but rather a direct fit of our atomic oxygen parametrisation (equation 7 in preprint) to the NRLMSIS model values for the lines-of-sight in a scan.*

Minor/specific comments

1. Lines 21-22: I am not quite sure what the authors mean by "[...] systematically analyses concentric atmospheric layers". I would say that the main intrinsic advantage of limb observation geometry is high vertical resolution. Onion peeling, on the other hand, is just one of the methods (and perhaps the simplest one), to do a limb retrieval. In my opinion, the fact that onion peeling retrieves the parameters of the outer layers first, and uses them as the "truth" to retrieve the inner layers is more of a flaw than an advantage.
Yes, we agree with that. We have rewritten the sentence to:
"Satellite observations using limb geometry improve the retrieval process by allowing systematic analysis of the atmosphere at different tangential altitudes."

2. Line 43: I would replace "non-spherical symmetric atmosphere" with "non-spherically symmetric atmosphere". More generally, the authors mention this lack of spherical symmetry quite a few times, but the first really clear explanation of what they mean is given in line 176: " [. . . ] variations in the vertical profiles across different longitudes and latitudes,

i.e. the spherical asymmetry of the atmosphere [. . . ]". I think this could be made clearer by either giving this explanation right from the beginning (i.e. line 43), or using some other term, like "horizontal structure of the atmosphere" instead of "spherical asymmetry". After line 43, I was left wondering whether the authors mean the horizontal structure, or do they mean that the oblateness of the Earth was accounted for in their calculations.

We changed the sentence at line 43 to: "Additionally, we account for spatial variations in atmospheric properties along the satellite's path and consider the impact of atmospheric winds."

And the sentence at line 128 to: "Consequently, the mission data is simulated using an atmosphere that varies with location, i.e. without assuming spherical symmetry"

Minor typos and suggestions

Here is the list of minor suggestions and typos that I have noticed. I do not expect point-by-point answers for these.

1. Line 59: Replace "5.2MHz (FHWM)" with "5.2MHz (FWHM)".
   Done.

2. 2. Line 184: "Without introduction of α, the retrieved profiles would correspond to some kind of averages over the regions traversed by the LOS." It seems to me that the benefit of correcting for spherical asymmetry in general is described here, and not an advantage of the particular parametrisation of the horizontal structure (i.e. the α angle). It would be good to make this clear.
   We changed the sentence to: "The introduction of such a correction has the additional advantage that vertical profiles can be obtained at arbitrary positions in the vicinity of the center of the combined scan. Without the correction the retrieved profiles would correspond to some kind of averages over the regions traversed by the LOS. These regions are larger than the ground track distance of the scans c.f. Figure 9."

3. 3. Line 238: "This retrieval resembles the example of a more accurate retrieval." Do the authors simply mean that this retrieval is an example of a more accurate retrieval? If it only "resembles the example", then where is that other example shown or described? The same applies for the similar statement in line 241.
   We changed the sentences to: "This retrieval is an example of a more accurate retrieval"

4. Penultimate line of the Figure 13 caption: replace "corresponds" with "correspond".
   Done.

---

## Author Comment (AC2)

Thank you very much for agreeing to review our paper and for your comments. We greatly appreciate your time!

Please find our replies in green below

Is the line width (sigma) in Eq.4 same as the shape parameter in Eq.6?
The sigma in Eq. 4 and Eq. 6 are different. We changed the symbol in Eq. 6 to $\kappa$. Thanks for catching this.

All simulations are from pencil-beam calculations. What would be the impact from antenna beam patterns, say 10 km field of view, on the simulated results and conclusions?
The assumption of a pencil like beam is justified by the mirror/telescope of 40 cm diameter which results in a 1 km FOV at the 100 km tangential point. A non-pencil like beam would have to be approximated in the forward model by a weighted average over several pencil-beam calculations. A wider beam will reduce the vertical resolution of the measurements, leading to less detailed retrieval results.

There are a number of single-sentence paragraphs. Not sure what is their purpose? If they are a key statement/point to make, the authors need to provide the supporting materials to help the reader better understand the claim. If they are a supporting sentence, the authors may consider to group them into the related paragraph.
We identified and corrected the two single-sentence paragraphs:

1. "For all the retrievals, the parameters for the atomic oxygen and temperature parametrisations as well as the spectral shifts converge well and within approximately 15 iterations (see supplementary materials for an example).", line 250, has been moved to line 231 which now reads:
"In total 31 retrievals have been performed. For all the retrievals, the parameters for the atomic oxygen and temperature parametrisations as well as the spectral shifts converge well and within approximately 15 iterations (see supplementary materials for an example)."

2. "Although the retrieval of winds was not the scope of this study, average winds along the LOS were retrieved and appear to be a good first approximation for capturing wind effects on the observed spectra.", line 276, has been moved a few lines before where we mention the average deviations:
"Using realistic THz receiver noise temperatures, atomic oxygen concentration were retrieved from 100km to 110km within ±15% average deviations and from 110km to 300km within ±3% average deviations. Temperatures were retrieved within ±3% average deviations from 100km to 175km. Although the retrieval of winds was not the scope of this study, average winds along the LOS were retrieved and appear to be a good first approximation for capturing wind effects on the observed spectra."

---

## Author Response (AR2)

Again, we would like to thank the reviewer for the time and thoughts invested. We really appreciate that.

At this point, I only have one further suggestion. In their response to my review, the authors provided a lot of interesting information about the wind retrievals. I respect their decision not to include all of that in the manuscript: these details are not absolutely necessary given the limited claims regarding wind retrieval results that the authors make in this work, and this omission is also understandable given that a more detailed wind treatment might be in the works. However, Figure 4 in the authors response shows an important (and positive!) result, namely that a simulated retrieval with zero winds was also performed as part of this study and it showed that the simplified wind treatment does not result in meaningful degradation of atomic oxygen retrieval quality. The authors could consider mentioning that in the manuscript, as this would reassure the readers that winds, which were included in the generation of the simulated data set for the main retrieval, can truly be regarded as being out of scope of this work (and perhaps addressed separately later).

As requested by the reviewer we have included Figure 4 in the author response to the manuscript. It has been merged into Figure 17 in the original manuscript.

To accommodate the figure, we also added a little text in the results section.

with LOS passing through such low altitudes. To access the impact of the simplified wind treatment in the retrievals (the approximation of LOS winds by Doppler shifts), the retrieval simulations were repeated for an atmosphere without winds. The resulting average deviations are also shown in Figure 17. The deviations are similar in both cases and the simplified wind treatment has little impact on the retrieval quality. Results from single scan retrievals without any corrections for spherical asymmetry can be found in the supplementary material.